# China's Engagement in Arctic Governance for Its Sustainable Development Based on International Law Perspective

**Jiayu Bai** [1,*] and **Kailei Zhu** [2]

1   School of Law, Nankai University, Tianjin 300350, China
2   School of International Affairs and Public Administration, Ocean University of China, Qingdao 266100, China
*   Correspondence: gracefulgl@hotmail.com

**Abstract:** With climate change, melting sea ice and snow in the Arctic increase the probability that states engage in activities there. The prosperity of Arctic activities serves as a reminder to the international community that the issue of Arctic governance must be prioritised to avoid Arctic unsustainable development. As a major stakeholder besides the Arctic states, it is necessary to study China's role in Arctic governance for its sustainable development to provide a reference for the diplomatic decision-making of other states. The paper selects international law as a perspective. It discusses China's engagement in the development of international law related to the theme from the global, regional, and bilateral/multilateral levels. At the global level, China's national role is characterized by engaging in global governance under the international rule of law for guarding the international order based on international law. At the regional level, China maintains the role of supporting and engaging in the Arctic Council, both before and after its establishment. At the bilateral and multilateral levels, China, as an Arctic stakeholder pursuing cooperation, has achieved many cooperation results. The paper holds that under China's national roles, China's engagement has affected the development of international law referred to the theme to some extent.

**Keywords:** Arctic governance; Arctic sustainable development; international law; China's role; Arctic cooperation





## 1. Introduction

Sea ice in the Arctic melts rapidly. On the basis of the U.S. Environmental Protection Agency, the sea ice area in the Arctic has highlighted a negative trend in recent decades [1]. Rapid Arctic environmental change affects the entire Earth's system as thawing permafrost ecosystems release greenhouse gases into the atmosphere, further accelerating global warming [2]. On the other side of the coin, along with climate warming, the melting of ice and snow in the Arctic has provided more opportunities for activities in navigation, mining, and tourism. With an increase in human activities in the Arctic, attention has been raised about competing for richness and economic advantages [3] (p. 1), which serves as a reminder to the international community that more attention must be focused on the issues of Arctic governance in order to avoid the accelerated Arctic $CO_2$ release caused by the Arctic unsustainable development, which will worsen the global climate and pose a threat to human survival.

The research on the Arctic sustainable development starts from the interpretation of the concept of sustainable development. As early as the 18th and 19th centuries, issues such as intrageneration and intergenerational equity, natural resource protection, and concern for the future have been discussed by European philosophers [4] (p. 367). In the early 20th century, scholars such as Vladimir Ivanovich Vernadsky [5] (pp. 167–176) and Kenneth E. Boulding [6] (pp. 3–14) began to research sustainable development. These theories reflect the common ground of attaching importance to the future of humankind and abandoning selfishness. Based on the accumulation of these theories, the World

Commission on Environment and Development published the report called *our common future* in 1987, which is considered the starting point for proposing the concept of sustainable development. It holds that sustainable development is development that meets the needs of the present without compromising the ability of future generations to meet their own needs [7]. Based on *our common future*, the international community has further explored the concept of sustainable development, including the World Bank [8] (p. 8) and economist Herman Daly [9] (pp. 39–53). In 1991, the book *Caring for the Earth—A Strategy for Sustainable Living* edited by the International Union for Conservation of Nature further proposed a concept of sustainable development: improving the quality of human life while living within the carrying capacity of supporting ecosystems [10] (p. 10). A sustainable society lives by several principles which contain a general principle and four standards. The general principle of "inspect and care for the community of life" is a basic principle that provides a moral basis for other principles. It requires governors to manage development so that it does not threaten the survival of other species or eliminate their habitats. The standard of improving the quality of human life emphasizes that economic growth is an important component of development, but it cannot be a goal in itself, nor can it go on indefinitely. The standard of conserving the Earth's vitality and diversity focuses on the conservation of life-support systems of ecological processes such as climate shaping, the biodiversity conservation, and the assurance of sustainable renewable resources uses. The standard of minimizing the depletion of non-renewable resources focuses on the conservation of non-renewable resources such as minerals, oil, gas, and coal. The standard of keeping within the earth's carrying capacity requires that policies that bring human numbers and lifestyles into balance with nature's capacity must be developed alongside technologies that enhance that capacity by careful management. The achievement of these standards depends on the establishment of a global alliance at the international level.

As an indispensable part of sustainable development, the sustainable development of the Arctic is closely involved in the implementation of sustainable development theory. From the global perspective, the sustainable development of the Arctic involves Arctic states and other stakeholders. There are many universal interests in the Arctic region, which links the interests of Arctic states and other stakeholders. Some Arctic sustainable development issues involve the common concern of humankind, such as climate change [11] (pp. 525–530) and the conservation of biological diversity [12] (pp. 171–180). In addition, international navigation [13] (pp. 770–783) and marine environmental protection [14] (p. 1) in the Arctic do not only matter for the Arctic states themselves. In order to put the theory of sustainable development into practice, the United Nations further divides the issue of sustainable development into specific goals which are called the United Nations Sustainable Development Goals (SDGs), such as the "SDGs 13—Take urgent action to combat climate change and its impacts" and the "SDGs 14—Conserve and sustainably use the oceans, seas and marine resources for sustainable development" [15]. From the regional perspective, the Arctic Council (AC), which takes the sustainable development and environmental protection in the Arctic as its own mission [16], has set the three themes of environmental protection (Arctic Climate, Healthy and Resilient Arctic Ecosystems, and Healthy Arctic Marine Environment), sustainable social and economic development (Sustainable Social Development and Sustainable Economic Development), and strengthening the AC (Knowledge and Communications and Stronger AC) as its goals in the 10-year plan 2021–2030 [17]. At the same time, any SDGs associated with the AC's mission and the projects and activities based on these SDGs are valued and implemented by the AC [18].

Governance is the general name of various ways in which all kinds of individuals manage their common affairs, and it is a continuous process of coordinating different interests to promote cooperation [19]. The governance process is dynamic. As the Arctic natural and political environments are constantly changing, Arctic governance for its sustainable development presents dynamic features. To facilitate an evaluation, it is necessary to select a representative, highly related indicator from the dynamic theme in order to solve a challenge that is difficult to assess owing to the dynamics of governance. Governance

and law are closely related [20] (p. 208). In global governance, the close relationship between international law and global governance proves it. Specific to the Arctic region, "the comprehensive governance model based on international law has benefited all Arctic states and the region as a whole" [21] (p. 1149). In view of the significant role of international law in Arctic governance for its sustainable development, it is reasonable to take up research from the perspective of international law.

On the evidence of the international law-making theory, international law is predominantly made by states and is often said to be a consent-based (or consensual) system [22]. Hence, when some proposals submitted by a state are adopted and embodied in the international treatyfinal version, it is generally considered that the state promotes the development of international law. Adoption is generally regarded as the substantive engagement of a state in international law-making. While, at the bilateral level, the substantive engagement is generally analysed from the results of the conclusion because the archives of concluding bilateral agreement are rarely open for the public. In addition, the contribution of codifying and progressively developing international law by diverse, non-binding, normatively worded instruments used in contemporary international relations by states and international organizations, called "soft law," cannot be ignored [23]. Furthermore, both bilateral treaties and multilateral treaties are structures in a complex architecture of legal instruments that make up international law. Neglecting the significance of both of them in this complex architecture impairs our descriptive acumen in that it leaves us with only a partial account of international law-making [24].

It is an undeniable fact that China is one of the major Arctic stakeholders besides the Arctic states [25] (p. 209). As for the geographical position, Chinese experts show maps of an expansive fifteenth-century empire that nearly touches the Arctic as proof of China's historical origin as that stakeholder [26] (p. 3). From the perspective of history, China joined the 1920 *Spitsbergen Treaty* as early as 1925. From the angle of climate change, as China is a Near-Arctic State [27], for the state's relatively northern latitudes [28] (p. 646), it is more affected by climate change from the Arctic compared with other non-Arctic states. A study shows that there is an association between spring Arctic sea ice concentration and Chinese summer rainfall [29] (p. 1). In terms of community communication, at the border, China has "Near-Arctic nationalities" [30]. The common well-being between Chinese Near-Arctic nationalities and the peoples in the Arctic region has established by community communication. Considering geographical location, history, climate change response, and community communication, China is one of the major Arctic stakeholders.

The authors pay attention to the ideas of ecological civilization and a community of life for man and nature proposed by China in recent years. These governance initiatives are highly consistent with the "respect and care for the community of life" of the sustainable development theory. Guided by these ideas, some research articles shed light on the increased credibility of the Chinese government's commitment to environmental protection since 2013 [31]. Adopting a four-wheel-driven approach that involves the government, enterprises, farmers, and academia, remarkable progress has been made in alleviating desertification and raising people's incomes in Kubuqi, the seventh largest desert in China [32]. Based on satellite data, China accounted for one-fourth of the globe's net increase in leaf area between 2000 and 2017 [33]. Accordingly, there are reasons to believe that China owns the ability and experience to conduct governance under the goal of sustainable development. In light of the close relationship between China and the two elements of the Arctic and sustainable development, it is rational to study China's engagement in Arctic governance for its sustainable development from the perspective of international law.

The authors intend to start with a detailed description of the development of international law for the related theme at the global, regional, multilateral, and bilateral levels. Then, on the basis of the international law-making theory, the authors apply the empirical study through the analysis of the proposal contents and their adoption to explore evidence regarding China's substantive engagement in the above-mentioned international law development at the global, regional, multilateral, and bilateral levels. These document

sources include the proposals of the Chinese representative, the adopted international law documents, the reports, and the resolutions of the conference of the parties and consensus documents. Then, according to the theory of National Role, National role is the general foreign policy behaviour of governments. National role is concluded from the policymakers' own definitions of the general kinds of decisions, commitments, rules, and actions suitable to their state, and of the functions, if any, their state should perform on a continuing basis in the international system or in subordinate regional systems.) [34] (p. 245). The authors apply the inductive study through the analysis of the official document contents to summarize the role of China at the above levels, because China's role guides a wide range of national practices, covering China's engagement in the international law development of Arctic governance for its sustainable development. In other words, the authors utilize China's role to figure out why China is willing to engage in the international law development of Arctic governance for its sustainable development and submit the proposals.

## 2. The Development of International Law and China's Engagement at the Global Level

At the global level, as one of the global regions, the Arctic region also faces challenges from many global issues, such as the maintenance of marine rights and interests, matters of common concern of humankind, including joint response to climate change and biodiversity conservation in whole aspects, and the navigation system in the open waters of the Arctic at relatively specific aspects. Since the solution of these global issues depends on the joint efforts of states in all regions, including the Arctic region, this requires that states, as the makers of international law, have the responsibility to formulate norms and systems to reflect their specific values and interests, any collective values or interests they may hold, or "the greater interests of humanity and planetary welfare" [35] (p. 2). Correspondingly, it is affirmed that the formation of a series of international laws and rules mentioned below has potential implications for Arctic governance. This is also the premise for the following discussion on China's dynamic engagement.

*2.1. The International Law Development of Arctic Governance for Its Sustainable Development at the Global Level*

2.1.1. The Field of Ocean Studies

The *Spitsbergen treaty* came into being earlier than the United Nations Convention on the Law of the Sea (UNCLOS). The *Spitsbergen treaty* is representative of this concept. The *Spitsbergen Archipelago* is a group of islands located in the Arctic Ocean. The legal status and the regime of the archipelago are governed by the unique *Spitsbergen Treaty* of 1920. Its emergence is of great significance for balancing the interests of Norway and other stakeholders in the archipelago. Before the conclusion of the *Spitsbergen treaty*, the interests of European states represented by Britain, Russia, and Norway on the archipelago of Spitsbergen were so complex that the sovereignty of the archipelago could not be clarified for a long time. After the end of World War I, the reset of the international order brought an opportunity to solve the issue of sovereignty over the archipelago. In this context, the fair system established by the *Spitsbergen treaty* helps solve the above issues in order to avoid contention between the contracting states for the rights and interests of the archipelago [36] (p. 2). The treaty also aimed to secure the economic interests of nationals from other states. This was achieved by including provisions on equal rights and non-discrimination in the most relevant economic activities [37] (p. 79). Under the treaty, the contracting parties enjoy the liberty of access and entrance within the Arctic as well as the right to carry out activities. Such rights include those of fishing and hunting in the territories specified in the *Spitsbergen Treaty* and in their territorial waters by ships and nationals of the contracting parties, the equal liberty of access and entry to the waters, fjords and ports of these territories, and the rights to carry out all maritime, industrial, mining, and commercial operations on a footing of absolute equality [38] (p. 266). Although there are details to be clarified, such as the conclusion of a convention on scientific investigation in accordance with Article 5, the treaty has smoothly existed for more than a century. At that time, the parties to the treaty

intended to solve the issue of territorial ownership, but objectively, the treaty provided favourable conditions for the parties to make rational and orderly use of the resources in the related area. It avoids the potential pre-emptive occupation and unsustainable development of related regional resources due to the dilemma of territorial ownership in a particular historical period. Specifically, the core intention of the contracting parties is to reach an agreement on how to maintain and exercise their rights over the contracting area so that the activities of the contracting parties to develop resources, including fishery resources, within the scope of application of the treaty are in an orderly state so as to ensure the peaceful use of the archipelago [39] (p. 159). The above provisions are advantageous to achieve the theoretical standard of sustainable development for maintaining the carrying capacity of the earth in specific regions of the Arctic.

The entry into force of UNCLOS has been widely recognized by States parties, including Arctic states and other Arctic stakeholders. There are customary international laws and general principles, including those related to the Arctic, such as the principle of sustainable development, which are applicable to the treatment of global international legal issues. Article 234 of UNCLOS is the only article directly related to the Arctic, which calls for marine environmental protection. The *Ilulissat Declaration* affirmed the role of UNCLOS in the governance of the marine environment, navigation, scientific research, and ice-covered areas, among other topics [40] (p. 817). UNCLOS protects the interests of humankind in the ocean by establishing legal order and provides the cornerstone for ocean studies [41] (p. 229). It proves that the parties of the convention are aware that all life on the earth is part of a huge interdependent system. Anthropogenic activities interference with the Arctic biosphere can affect the whole. Accordingly, it provides a basic legal framework for the follow-up Arctic navigation, joint response to climate change [42] (pp. 406–409), biodiversity conservation [43] (pp. 188–191), and other issues. This approach has benefited the present and future generations of Arctic residents who are part of the community of life.

### 2.1.2. The Field of Arctic Navigation

Economic growth is an important component of development. Although people set different goals for development, some goals are actually universal and represent universal interests, such as the Arctic navigation mentioned above. The *International code of safety for ships operating in polar waters* (PC) is a separate set of legally binding rules [44] (p. 677), which is used to guide the benign development of Arctic navigation. It balances economic development and environment protection when navigating in Arctic ice-covered waters, makes our life better in the economic and environmental fields, and makes for ensuring improvement of the quality of human life. However, the PC has not solved all international legal issues surrounding polar shipping. It is not applicable to state owned or operated vessels, smaller vessels, leisure boats, and fishing boats [45] (p. 368) and also fails to address issues of invasive species [46] (p. 176). Some sea ice areas which pose a structural risk to ships have been excluded in the PC. For instance, these areas are the North Atlantic Ocean to part of the Norwegian Sea along the shore of Norway and the adjacent part of the Barents Sea to the Kola Peninsula in Russia [47] (p. 219). It is worth affirming that the PC adapts to the latest polar navigation demand and natural environment by constantly improving itself. For example, the International Maritime Organization's (IMO) Marine Environment Protection Committee (MEPC) has adopted a resolution prohibiting the use of heavy fuel oil (HFO) in the Arctic, although certain exemptions mean that a complete ban in the Arctic is still years away [48] (P. 274). Especially in the context of the COVID-19 global pandemic, the IMO provided practical suggestions and guidance [49].

### 2.1.3. The Field of Joint Response to Climate Change

The increase in Arctic navigation is due to the melting of ice and snow in the context of Arctic climate change. However, the melting of ice and snow may also exacerbate the loss of biodiversity in the Arctic, which relies on ice and snow for survival, thus increasing the risk

of unsustainable development faced by humankind. In response to this concern, the U.N. General Assembly established the Intergovernmental Negotiating Committee in December 1990 [50] and adopted the *U.N. Framework Convention on Climate Change* (UNFCCC) on 9 May 1992. It is a considerable achievement to reach an agreement recognized by more than 140 states with different interests in such a limited time [51] (p. 454). It is necessary to point out that the Convention contains only a vague set of commitments regarding stabilization and no commitment to all on reductions. The *Kyoto protocol* adopted in 1997 is an attempt to implement specific measures after the Convention. It has set up a mandatory emission reduction system, but the effect is limited [52]. Some developed states listed in Annex I argue that it would affect domestic employment and cause economic losses [53] (p. 46). The key to curbing the serious threat of global warming is the investment in research, new technology, and tax incentives to promote voluntary reductions, as opposed to the imposition of mandatory regulatory target levels of emissions [53] (p. 46). For the sake of optimizing the problems existing in the *Kyoto Protocol*, based on the framework, more specific agreements are gradually produced to implement climate change response measures. Accordingly, the Warsaw climate change conference held at the end of 2013 first proposed the concept of Intended Nationally Determined Contributions (INDCs) [54] and further clarified its content and form at the Lima climate change conference in 2014 [55]. The *Paris Agreement* was a milestone, and global economies agreed to make every attempt [56] (p. 11881) to ensure that the planet's temperature does not rise above 2 °C. The achievement of this ambitious goal relies on nationally determined contributions (NDCs) [57] (p. e93). After the ratification and entry into force of the *Paris Agreement*, the INDCs have be transformed into the official NDCs. From UNFCCC to the *Paris Agreement*, these efforts help protect the ecological processes that shape the climate to protect the life support system that maintains the vitality of the earth and reach the theoretical sustainable development standard of conserving the earth's vitality and diversity.

The climate change response in the Arctic is facing challenges related to the design of earth system models and the implementation of the *Paris Agreement*. From the perspective of carbon release, most earth system models do not consider the process by which permafrost may lead to carbon release. Therefore, the *Paris Agreement* needs to assess the specific circumstances of the Arctic. In addition, climate governance depends on governance mechanisms, knowledge, and funds, which entail strict requirements for all states [58] (p. 2). To meet these challenges, the *Paris Agreement* is making corresponding adjustments. Two international carbon markets named the framework of various approaches, and the new market mechanisms were introduced recently in the light of Art. 6 of the *Paris Agreement* that affect the post-2020 climate regime [59] (p. 21).

### 2.1.4. The Field of Biological Diversity Conservation

Climate change has caused habitat change, which has broken the balance of ecosystems and threatened the diversity of genes, species, and ecosystems. Hence, Arctic biodiversity also deserves global attention. The *Convention on Biological Diversity* (CBD) provides solutions to biodiversity challenges within national jurisdictions [60] (p. 3295). However, the scope of the application of the CBD does not include areas beyond national jurisdiction, so it needs to be supplemented by other agreements. The *international legally binding instrument under UNCLOS on the conservation and sustainable use of marine biological diversity of areas beyond national jurisdiction* (the BBNJ Agreement) fills the gap. By stressing universal participation, the BBNJ Agreement contributes to sustainable development [61], reflecting the consideration of the common wellbeing of humankind [62]. Biodiversity conservation in the Arctic is closely related to the diversity of the earth in the standard of sustainable development. The ecology of the Arctic is fragile, and the formulation of international laws regulating the utilization of biological resources in the Arctic is conducive to preventing the uncontrolled development of its biological resources. After four preparatory committee conferences and five intergovernmental negotiations, the agreement has been reached.

During the negotiation, the most common consensus is related to the theme of capacity-building and technology transfer, which will be described in detail below.

### 2.2. China's Arctic Substantive Engagement in the International Law Development of Arctic Governance for Its Sustainable Development at the Global Level

#### 2.2.1. The Field of Ocean Studies

In the light of the international law-making theory, it is necessary for the authors to find China's dynamic engagement results of the international law development related to the paper's theme by examining the adoption of China's proposals in the international law-making in the above treaties. As one of the parties to the *Spitsbergen treaty*, China (the Beiyang government) was first invited by France and joined the treaty on 1 July 1925. Due to the shortage of data about the negotiating or signing the *Spitsbergen treaty*, the authors do not find substantive engagement of China in the period of negotiating or signing the *Spitsbergen treaty*. China built a permanent research station named the Arctic Yellow River Station on Svalbard Island in 2004, which was permitted by the treaty [63] (p. 11). These scientific research practices proved that the more China conducts scientific research activities in accordance with the *Spitsbergen treaty*, the clearer the intention to establish long-term legal relations in accordance with the *Spitsbergen treaty* would be reflected [39] (p. 159). The AC's sustainable development goal of healthy and resilient Arctic ecosystems and the realization of the SDGs 14 all depend on the best scientific evidence. China has contributed to the collection of scientific evidence within the treaty area. Such scientific evidence could provide preparations for a new convention conclusion on scientific investigation in accordance with Article 5 of the *Spitsbergen treaty*.

Compared with the shortage of data about negotiating or signing the *Spitsbergen treaty*, the negotiation archives of UNCLOS are relatively complete. Since the restoration of the lawful seat of the People's Republic of China in the United Nations, Chinese representatives have actively engaged in the formulation of UNCLOS. China's relevant proposals can be roughly divided into four categories. China stresses marine environmental protection, sharing in scientific research, freedom of navigation, and sustainable development in resource exploitation [64] (pp. 79–99). As a newcomer on the international stage, the negotiation conference did not adopt many proposals at the beginning of the negotiation. With the accumulation of diplomatic experience along with China's reform and opening up, the substantive achievements of China's participation in the conference have gradually increased [65] (p. 33). Some of China's statements have been considered into the final version of UNCLOS, such as the transfer of scientific achievements from developed states to developing states, the purpose of peaceful exploitation of seabed resources, equal access to seabed resources regardless of the size of states, etc. (All China's proposals are quoted from a series of books called Documents of the Chinese delegation to relevant United Nations meetings (in Chinese, published by the People's Publishing House). Except for 1973, this series is published every six months from 1974 to 1982.). In addition, China stated that the international seabed area and its resources are the common heritage of mankind and that the activities of exploration and mining in deep seabed areas need to be managed by international law [66] (p. 276). In line with the international law-making theory, these substantive engagements shows China's contribution to the development of the international law related to the paper's theme, although China did not put forward very specific ideas on the Arctic at that time [67] (p. 2). The above proposals on marine environmental protection have encouraged safety at sea, prevention of marine pollution, and cooperation to improve knowledge of the Arctic marine environment. It advanced the realization of the AC's sustainable development goal of a healthy Arctic marine environment. At the same time, since the above proposal also provides effective legislative advice on issues such as deep-sea mining including the area in the Arctic, it ultimately supports the implementation of the SDGs 14.

2.2.2. The Field of Navigation

Due to the remoteness and complexity of the Arctic region, it is necessary to formulate a separate set of legally binding rules to guide navigation. Until 2014, China began to put forward relevant suggestions on PC legislation. China's rising engagement in the law-making of navigation regulations in polar waters can be observed through the number of proposals by China. According to the international law-making theory, the authors mainly focus on the following four results of China's substantive engagement).

China actively engaged in the IMO law-making activities related to Arctic navigation and had a direct effect on the PC text. It resolves the contradiction that ships with lower oil pollution risk have more stringent structural requirements, while ships facing higher oil pollution risk enjoy less stringent requirements. China and the Republic of Korea jointly submitted a proposal on the content of environmental protection in the draft PC at the 68th MEPC meeting in 2015 [68]. The General Assembly agreed in principle with the proposed modifications to regulations 1.2.2 [69].

China's proposals are also concerned with the transition period and promotion of the crew training for Arctic navigation by engaging in PC-related law-making activities indirectly. China submitted a proposal in 2014 [70] on the content of training requirements for personnel on ships operating in polar waters before the draft PC was developed or finalized at the first Sub-Committee on Human Element, Training and Watchkeeping (HTW) meeting in 2013. The decision of the General Assembly to consider it at an appropriate time [71] affirmed the above views of the Chinese representative to a certain extent. As the provision was used as an interim provision before PC took effect, the revision promoted the smooth transition of the rules and supplied a good reference for the formation of PC in formulating the above terms for crew training.

In addition, by sharing information and actively completing the performance of the Sub-Committee on Pollution Prevention and Response (PPR) working group [72] (p. 9), China has made preliminary preparations for the formulation of rules around controlling black carbon emissions in Arctic navigation. In accordance with the relevant requirements of the instructions of MEPC 74 in 2019 and the communication group of the PPR 7 in 2020, China carried out research related to black carbon emission control and shared information on its ongoing black carbon project in 2021 [73–75].

It is worth noting that in addition to hard laws such as PC and others regulating black carbon emissions from ships in polar waters, soft law engagement activities attended by China via the IMO are also more frequent. China has published the *Guidance on the Prevention and Control of COVID-19 on Board* via the IMO to the world since 2020 [76]. Guided by the circular, the infection incident on the container vessel Gjertrud Maersk was successfully handled, and the ship was able to resume navigation in time [77].

By the international law-making theory, the above substantive engagements reflect in the types of adopted proposals and have positive effects on the international law development involving the paper's theme. They are advantageous to promote innovative, sustainable, and low-emission technologies and maintain the balance between economic development and environmental protection in the Arctic so as to achieve the AC's sustainable development goal of sustainable economic development. Meanwhile, the implementation of these measures proposed by China is valid for fulfilling the SDGs 14.

2.2.3. The Field of Joint Response to Climate Change

It is such an urgent fact that the Arctic is more impacted by global warming than any other place in the world [78]. Hence, it is pressing to speed up the formulation of relevant international laws to deal with the sustainable development issues caused by the increasingly serious climate warming in the Arctic. China has actively engaged in law-making activities in this field. In regard to the international law-making theory, the following is China's substantive engagement combed by the authors through the negotiation documents, namely the adoption of the proposals.

During the Intergovernmental negotiations of the UNFCCC, China put forth the idea of formulating a "Framework Convention", which is supported by the group of 77 members, and the conference ultimately adopted this idea [79] (p. 208). China has also put forward a complete proposal on the draft convention during the negotiation process. This is the first time that China has provided a complete text of the draft convention in multilateral treaty negotiations [79] (p. 207). The draft consists of 26 articles. The expression in the draft that "the international community has common but shared responsibilities in dealing with climate change" is the prototype of the "common but differentiated responsibilities" established by the follow-up conference of the parties [79] (p. 211). China stressed that the joint activities of developed and developing parties are different from joint implementation [80]. These joint activities should cooperate with and support national priority areas and strategies for sustainable development and should promote technical cooperation, including technology transfer and capacity building. This declaration is reflected in Article 4.5 of the adopted text [81].

In the negotiation stage of the *Paris Agreement*, the scope of provisions related to NDCs is diverse, involving mitigation, adaptation, finance, technology development and transfer, capacity-building, and transparency, among others. China has put forward some views on NDCs systems similar to those of Arctic states. Both China and Iceland agreed in the submission documents about views regarding the process and outcome of the 2015 agreement negotiation that any agreement that needs to be finalized in 2015 should include the transparency of mitigation, adaptation, means of implementation, action, and support and should be legally binding at the international level [82,83]. Canada also stressed in the submission that the agreement should contain key themes similar to those above [84]. In fact, the official text of the *Paris Agreement* is legally binding and does contain the above themes [85]. In addition, based on the *Paris Agreement*, China has created several INDCs according to their actual situation. China constantly puts forward new emission reduction targets and makes efforts to fulfil its commitments in the implementation of the *Paris Agreement* (See Table 1). China made its first commitment to reduce carbon emissions in 2009 and fulfilled it two years ahead of schedule [86]. In the Paris climate conference of 2015, China pledged to peak $CO_2$ emissions by around 2030 and strive to achieve it as soon as possible. In 2020, China refreshed the above commitments, that is, to have a $CO_2$ emissions peak before 2030 and achieve carbon neutrality by 2060 [87].

**Table 1.** Main targets for 2030 in China's First NDC (2016) and its revised version (2021) as well as the progress to date as of the end of 2020 [88]).

| Indicators | Targets for 2030 | | Progress in 2020 |
|---|---|---|---|
| | First NDC (2016) | Revised NDC (2021) | |
| peaking $CO_2$ emissions | "around 2030" (and "making best efforts to peak early") | "before 2030" (and "achieve carbon neutrality before 2060") | around 80% of China's emissions "having peaked" or "expected to peak before 2025" |
| $CO_2$ intensity reduction (compared to 2005) | 60–65% | >65% | 48.4% |
| non-fossil share in primary energy mix | around 20% | around 25% | 15.9% |
| forest stock volume increase (compared to 2005) | around 4.5 billion cubic metres | 6 billion cubic metres | 5.1 billion cubic metres |
| installed capacity of wind and solar power | – | >1200 GW | 534 GW |

By the international law-making theory, this substantive participation of China denotes China's promotion to the development of international law concerning the paper's theme. These adopted proposals require Arctic states and other stakeholders to monitor, assess,

and highlight the impacts of climate change in the Arctic for the purpose of the SDGs 13 and the AC's sustainable development goals on climate. Meanwhile, these opinions support the adoption of stronger global measures to reduce greenhouse gases and short-lived climate pollutants and enhances the Arctic's adaptability and resilience to climate change.

### 2.2.4. The Field of Biological Diversity Conservation

The emergence and development of the CBD and the BBNJ Agreement indicate that global governance in the biological diversity field has attracted much attention. China has been actively participating in this developing field of biodiversity conservation since the formation of the CBD draft. In the fifth negotiating session of the Intergovernmental Negotiating Committee in 1991, the Committee quoted a Chinese proposal at the third session of the Preparatory Committee for the United Nations Conference on Environment and Development in 1991 as a reference for the legislative wording of Article 16 (acquisition and transfer of technology) of the CBD draft [89]. The access to and transfer of the technology system described in this article is retained in the effective version. This substantive engagement (that is, the adopted proposals by the theory of international law-making) is the impetus to the development of international law concerning the paper's theme. In detail, both the establishment and management of protected areas and the implementation of environmental impact assessments (EIAs) are inseparable from professional technology. However, the fact is that there is a gap in the capacity and technology of developing states compared with developed states. The shortage of technology in developing states is not conducive to the global conservation of biodiversity. Accordingly, it is necessary to pay special attention to these shortages to achieve worldwide cooperation. The clause of acquisition and transfer of technology contributed by China is conducive to enhancing developing states' understanding and mastery of biodiversity conservation technologies in the Arctic and enhancing their ability to engage in Arctic affairs.

In recent years, China has actively hosted the Conference of the Parties. During the 15th meeting of the Conference of the Parties to the CBD hosted by China, the *Kunming Declaration* was adopted. The significance of the *Kunming declaration* to the Arctic is that it provides guidance for Arctic states to continue to protect biodiversity under national jurisdiction after 2020 [90]. The endeavours of China are influential in providing a platform for the discussion of biodiversity solutions and reflect China's contributions to issues in the conservation of biodiversity in the Arctic. The above suggestions and efforts prove that China owns the ability to promote the "SDGs 15- Department of Economic and Social Affairs" and the AC's sustainable development goal of healthy and resilient Arctic ecosystems.

China has actively engaged in the negotiation of the BBNJ Agreement. In currently published materials(as of the fifth resumed intergovernmental negotiating conferences) related to the agreement negotiation, consensus was reached on the four core issues involved in the agreement mentioned above. Although the BBNJ Agreement has been reached, the complete documents of the resumed fifth intergovernmental conference are unavailable as of March 19, 2023. Accordingly, the authors attempt to compare these consensuses recorded in the report of the conference with China's related proposals which are from before the resumed fifth intergovernmental conference. In regard to the international law-making theory, the authors seek to anatomize how China's substantive engagement in the negotiation of this agreement that takes these coincidence points as the carrier promotes the development of international law related to the theme.

First, as mentioned earlier, capacity-building and the transfer of marine technology are of great significance to the BBNJ Agreement. The effective implementation of the agreement depends on the premise that parties are equipped with effective capacity-building and technology transfer. The views on the types [91,92], modalities [91–93] (p. 45), cooperation [92–96] (pp. 32, 46, 79), funding [94,97], and supervision [92,94,97–99] have been mutually confirmed with the consensuses in the report of the president of the General Assembly. These facts are of great significance to the sustainable use of biological resources

in the Arctic. Specifically, a flexible list of categories or types helps to better identify the actual needs of parties for biodiversity conservation technologies in relevant Arctic regions. The establishment of the information exchange mechanism and cooperation on the modalities and implementation can facilitate the promotion of the mastery and efficiency of technologies related to human well-being for the conservation of biodiversity in the Arctic areas beyond national jurisdiction. The agreed funding acquisition and supervision stipulation help provide sufficient funds for Arctic technology transfer and strengthen the legitimacy of the process.

Second, EIAs and area-based management tools (ABMTs) are two primary means of implementation. The agreed guiding principles and approaches of EIAs [91–93] (pp. 72, 95) require relevant subjects to adopt a flexible list of categories [91–93,95] (p. 75) and ensure that the details of the contents of the EIAs report are appropriate [91–93] (p. 140). The assessment of the establishment of marine protected areas as one of the ABMTs [92,93,100] (pp. 15, 29, 65, 94) should be based on broad participation with stakeholders [92,95,101] (p. 70). The views are consistent with the consensus and are meaningful to Arctic biodiversity conservation. For the EIAs, its application in the Arctic under the BBNJ Agreement is worth making any possible adjustments to promote the proposed activities. Meanwhile, it is in favour of preventing or mitigating possible marine environmental pollution or harmful changes. Ultimately, it would help facilitate the implementation of the proposed activities. For the ABMTs, the Arctic ecosystem's natural biochemical processes are slowed by cold, extreme seasonal variations in light and extensive ice cover. Accordingly, it is necessary to establish marine protected areas in relevant places in the future. The consensus to ensure the broad engagement of Arctic stakeholders is favourable to the implementation of the agreement.

Finally, the purpose of the convention, as described in the title of the BBNJ Agreement, can be achieved by meeting the above premises and using the above means. After unanimously approving the inclusion of the definition of marine genetic resources in the text [92,93,97–99] (p. 64), the adoption of the sharing of nonmonetary benefits [92,93,97,98] (p. 63) was widely recognized by the conference and Chinese representatives. As the majority of marine genetic resources (MGRs), including those in the polar regions, are located in waters that are beyond all national jurisdictions [102] (p. 273), how to sustainably use MGRs to benefit humankind is an issue that needs rational thinking in the future. MGRs are the biological building blocks for biodiversity in all of these areas [103] (p. 1). The necessity of including MGRs in the scope of application of the BBNJ Agreement has been recognized by consensus. In addition, considering the conditions for large-scale commercialization are not yet mature, giving priority to nonmonetary benefit-sharing rather than monetary benefit-sharing, such as convenient access to samples, information exchange, technology transfer, and capacity-building, is positive for encouraging the enthusiasm of researchers.

By comparing the views of China's proposals with the consensus reached at the Conference, it can be observed that almost all of the consensus reached there can be found in China's proposals. Chinese views are consistent with the consensus of the General Assembly. They are conducive to the entry into force and improvement of the BBNJ Agreement and in turn promote the development of international law for the Arctic biodiversity beyond national jurisdiction. This is favourable to the conservation of Arctic biodiversity, ecosystems, and species' habitats in order to achieve the SDGs 14 and the AC's sustainable development goal of healthy and resilient Arctic ecosystems.

*2.3. Implications of China's Arctic Engagement in the International Law Development of Arctic Governance for Its Sustainable Development at the Global Level*

As mentioned above, international law at the global level has potential implications for Arctic governance. Moreover, the national role in diplomacy guides the process of state engagement in the formation of international law. Accordingly, the paper intends to summarize China's evolving diplomatic role since 1971 to explore the relationship between

that role and engagement in the formulation of the international legal framework at the global level.

2.3.1. Discussion on China's Role Assessed by Its Official Documents at the Global Level

The national role in diplomacy has guiding significance for the state to engage in the development of international law. Meanwhile, since international law is also applicable to the Arctic region, and these international laws are related to the Arctic governance for its sustainable development, the role of the state is also related to the Arctic governance for sustainable development. Based on the above analyses, the paper summarizes China's implications by observing the national role and explaining China's engagement as well as analysing the impact on the development of international law for Arctic governance. The text used to analyse the role of this is based on the documents of the Chinese representatives presented in the United Nations General Assembly, which spans from the early 1970s to the late 1980s. At this stage, the policy process represented by the Third United Nations Conference on the Law of the Sea has been taking shape ever since [104] (p. 9). The authors observed that China has repeatedly emphasized the political and economic aspects of the "development of developing states" in the United Nations general debate. Specifically, China emphasized its status as a developing state with a backward economy [105] (pp. 5–6), reminded the world of the importance of providing assistance to the developing states in the greatest need [106], and advocated for uniting with third world states [107]. China also pointed out that the establishment of a new international economic order is what developing states require [108], proposed the help of developed states to developing states [109], and promised to give full play to China's own contribution [110]. China's foreign policies also pointed out the reality of poverty in the emerging stages of developing states [111] and advocated strengthening cooperation with the third world states [112]. In addition, China commenced to stress that the settlement of international disputes depends on the international law since 1980. Obviously, China emphasized its role as a developing state in need of the mutual assistance of the third world and the care of developed states at that stage on the premise of respecting international law.

From the 1990s to 2010, China paid more attention to its own development with changes in the international circumstances of peace and development [113]. The development of the national role has affected China's engagement in the development of international law at the global level, which also applies to the Arctic region. In this context, states all over the world have gradually been absorbed in developing their own economies and have begun to commit to dealing with some of the environmental and climate issues caused by national development. China consistently expressed respect to international law in the United Nations general debate. Through the observation of China's foreign policies, the authors find that the orientation of foreign policy remained basically stable for more than 20 years. Specifically, the foreign policies of this period emphasized adhering to the five principles of peaceful coexistence [114] (p. 1437), paying attention to its own development [115] (p. 1346), and striving to assume responsibilities and obligations in international affairs that are commensurate with its own capabilities [116] (p. 237) at the right time. Judging from China's diplomatic speeches at international occasions, at the United Nations General Assembly during this period, China conveyed that China is still a developing state and pledged to participate in international conferences related to the environment and climate [117,118]. Since the United Nations General Assembly incorporated for the first time the item entitled "The rule of Law at the national and international levels" into the agenda of the Sixth Committee in 2006, China started to express its views about international rule of law via this occasion. During that period, China repeatedly reaffirmed the strengthening of the international rule of law by upholding the authority of the United Nations Charter and the fundamental principles of international law, improving international legislation, adhering to the uniform application of international law, and promoting democratization in international relations [119]. Accordingly, this evidence shows that China emphasized its role in diplomacy in view of its ability at that stage. It

emphasizes avoidance of conflict with the great powers subject to international law at the national and international levels and strives to strengthen its economy and technological base.

Over the past decade, with enhanced comprehensive national strength in the previous period, China began to pay more attention to the building of the global governance system [120] followed by the new trend of the multilevel global governance system [121] (p. 94). It stressed upholding the international system with the United Nations at its core [122], advocated cooperation [123], and called for maintaining the common well-being of humankind [124] and boosting a community with a shared future for humankind [125–127]. These years, China gradually initiated the international order based on international law and expressed this initiative on various occasions under the United Nations, such as the United Nations Security Council Open Debate on Peacebuilding and Sustaining Peace. These diplomatic initiatives coincide with the general principle of respecting and caring for the community of life in the theory of sustainable development. Their realization is inseparable from the joint efforts of all states worldwide. Accordingly, China's role during the period from 2011 to present is characterized by engaging in global governance under the international rule of law for guarding the international order based on international law (See Table 2).

**Table 2.** China's Arctic dynamic engagement in the international law development of Arctic governance for its sustainable development at the global level under the national role.

| Period | International Law Context | China's National Role | Proposals Key Words | Adoption Outcome |
|---|---|---|---|---|
| 1970s to the 1980s | third United Nations Conference on the Law of the Sea | a developing state in need of the mutual assistance of the third world and the care of developed states at that stage on the premise of respecting the international law | marine environment; marine scientific research; right of navigation; marine resource development | many adopted proposals assessed under the condition of relatively complete archives |
| 1990s to 2010 | Convention on Biological Diversity and United Nations Framework Convention on Climate Change negotiation | a developing state carrying out diplomacy in view of its ability | acquisition and transfer of technology; common but differentiated responsibilities; joint activities | a few adopted proposals assessed under the condition of limited archive |
| 2011 to present | Polar Code and the Marine Biodiversity of Areas Beyond National Jurisdiction Agreement negotiation | a developing state engaging in global governance under the international rule of law for guarding the international order based on international law | training requirements for personnel on board ships operating in polar waters; ships structural requirements under oil pollution risk; black carbon emissions; marine genetic resources; area-based management tools; environmental impact assessments; capacity-building and the transfer of marine technology | a few adopted proposals assessed under the condition of relatively complete archives |

### 2.3.2. Discussion on China's Arctic Engagement under Its Role at the Global Level

From the early 1970s to the late 1980s, since the restoration of the People's Republic of China's lawful seat in the United Nations, China has shown a high level of enthusiasm and has been very active in sessions and debates with developed states for its friends in the third world [128] (p. 215). Accordingly, the national role during this period provides a reasonable explanation for China's engagement in the formation of UNCLOS. Moreover, the

contents of these proposals objectively follow China's national role during this period. The peaceful use of the ocean, the contents of technology transfer, fair systems, and licensing systems emphasized by the Chinese representative in the proposals can, on the one hand, promote the balance of the benefits of the exploration and mining in the deep-sea areas between developed and developing states. On the other hand, as UNCLOS contains approximately 25 references to the need for assistance in developing states and accounting for their concerns, these China's proposals help enhance the capacity of developing states.

The shifting international development trends at the stage from the 1990s to 2010 generate new diplomatic needs, which is the main reason for the conclusion of the new international laws such as the CBD and the UNFCCC. The emergence of these international laws has further promoted the legal framework centred on UNCLOS related to the theme. China fulfils its commitment, which forms the characteristics of China's role in this period, to participate in international conferences related to the environment and climate. Affected by the national role, some climate and biodiversity conservation proposals are in line with China's own strength and no longer show aggressive ambition. Regardless of the principle of common but differentiated responsibilities or the technology transfer of biodiversity conservation from developed states to those developing states, China's purpose is to seek a competent position in international law and expect to assume international responsibilities and obligations commensurate with their own capabilities.

The currently complex international situation poses challenges to the status of the international legal framework at the global level. The new development of international law in this field, particularly the emergence of PC, provides more accurate guidance for the Arctic navigation. China witnessed its role with the characteristics of active rather than passive participation in global governance in this period.. Such characteristics persuasively interpret why China seeks to introduce the domestic idea of ecological civilization to the CBD and to make new INDCs for the *Paris Agreement*. Such characteristics also explain why many of the views in China's proposals are consistent with the consensus of the BBNJ Agreement negotiating conference. Based on the characteristic of respect for the international rule of law reflected by China's role, it is China's commencement to participate in the formulation of the PC by taking advantage of its membership in the IMO that makes China's participation in the Arctic region based on the global international rule of law.

## 3. The Development of International Law and China's Engagement at the Regional Level

The Arctic, as a region for strategic competition, has seized the world's attention, but it is also necessary to ensure the rule of law so that it remains a region free of conflict where states act responsibly [129]. At the regional level, the international law for Arctic governance focuses on common Arctic issues, particularly on issues of sustainable development and environmental protection in the Arctic. These issues are closely related to the mission of the AC. In the AC, Arctic states and other Arctic stakeholders have different rights and obligations, similar to the status of "member states" and "observer states," respectively. Accordingly, international law for Arctic governance at the regional level refers to the provision of public goods to the Arctic region by Arctic states and other Arctic stakeholders through forums with the AC as the core, on the premise of being endowed with different rights and obligations.

### 3.1. The International Law Development of Arctic Governance for Its Sustainable Development at the Regional Level

Global regulations can provide a basic framework but to some extent may be weak in meeting the special needs of Arctic governance. Therefore, in recent years, regional treaties specifically applicable to the Arctic have been successively adopted. Among them, the most important are the *Agreement on Cooperation on Aeronautical and Maritime Search and Rescue in the Arctic* (SARA), the *Agreement on Cooperation on Marine Oil Pollution Preparedness and Response in the Arctic* (MOSPA), and the *Agreement on Enhancing International Arctic Scientific Cooperation* (EIASC). On 20 May 2021, foreign ministers of the eight Arctic states as

permanent members of the AC adopted a first 10-year strategic plan for the region [130]. It is the AC's first-ever Strategic Plan that reflects the shared values, goals, and joint aspirations of the Arctic states and Indigenous Permanent Participants, which will guide the Council's work for the next decade. The strategic plan covers almost all concerned fields in today's Arctic region. Therefore, it can be inferred that it will play an important guiding role in today's Arctic governance. Through this declaration and plan, the council can act as a promoter to make the Arctic a region of peace, stability, and constructive cooperation, which will allow for stakeholders, including China, to engage in Arctic affairs. This plan allows stakeholders, including China, to further engage in Arctic affairs to make the Arctic a peaceful, stable, and constructive region of cooperation.

The formation of the abovementioned regional treaties reflects an important difference between regional and global international law for Arctic governance. The most influential platforms generated by these treaties at the regional level are relatively centralized and dominated by Arctic states. These important regional treaties were reached through the platform of the AC. Although the AC is a semigovernmental forum, it has become the core institution for regional governance and cooperation in the Arctic [131] (p. 780). Just as human societies are interdependent, and future generations are affected by our present actions, the world of nature is increasingly dominated by our behaviour. The 10-year plan and a series of agreements adopted by the AC require the members of the AC to manage the Arctic climate, environment, economy, scientific activities, local knowledge, and other fields [129] so that human activities in the above fields in the Arctic region do not threaten the survival of other species or eliminate their habitats. It is a practice of care for other people and other forms of life and avails ourselves of meeting the general requirements of respecting and caring for the community of life in the theory of sustainable development.

What deserves deep attention is that some recent changes in the AC affect the normal operation of the AC mechanism. Seven of the eight states that make up the AC temporarily paused most of their cooperation due to the situation in Ukraine [132] in March 2022 [133]. It seems to expose the myth of "Arctic exceptionalism"—the idea that the Arctic is impervious to, or at least isolated from, the conflicts plaguing the rest of the world [134]. With the temporary pause of most work of the AC, the impact is complex. Scientists worry that "a range of research priorities, including the monitoring of wildfires, thawing permafrost, and methane emissions could be disrupted by an extended interruption in data collection and sharing" [135]. Considering the enduring value of the AC for the above work and the commitment of the AC members when they joined, since June of the same year, seven states except Russia intend to implement a limited resumption of their work in the AC in projects that do not involve the participation of Russia [136]. Nevertheless, given that Russia accounts for 50% of the Arctic landmass, it seems difficult to promote the operation of these projects without Russia. Since the issues caused by climate change are not those that can be suspended by human beings [137], although the impact of the temporary pause on the AC cannot be predicted in a short amount of time, it is obviously not conducive for the AC to achieve its purpose if it is paused for a long time. As no similar situation was expected during the formulation of the AC, no alternate plan can be adopted [138]. It is necessary to observe the impact of the conflict on the AC for a long time.

*3.2. China's Arctic Substantive Engagement in the International Law Development of Arctic Governance for Its Sustainable Development at the Regional Level*

China is not a party to SARA, MOSPA, and EIASC, but it would still like to respect these treaties and help with implementing them in practice. The evidence reflecting in *China's Arctic policy* said that China respects the agreements adopted by the AC [27]. Such recognition is the premise for China to engage in Arctic governance for its sustainable development at the regional level via the AC from the perspective of international law. By the international law-making theory, the substantive practices below explain China's efforts to promote the regional international law development concerning the paper's theme via the AC.

Before the formal establishment of the AC on 19 September 1996 [139], China had tried to participate in activities at the Arctic regional level as a stakeholder. As early as 8 November 1991, 11 regions from 9 northern states [140] established the Northern Forum (NF) in the United States. It became an observer for the AC in 1998 [141]. As one of the founding members, the then governor of the Heilongjiang Province of China served as the vice chairman of the NF [142,143]. On 23 April 1996 [144], China joined the International Arctic Science Committee [145] (p. 26), which also became an AC observer in 1998 [146]. Although the authors have not collected more archives from that period about China's engagement in the Arctic governance for its sustainable development at the regional level in this period, a series of activities related to China's engagement as a member of the Arctic regional organizations have laid a foundation for China to engage in the AC and carry out activities.

After the establishment of the AC and before China became its official observer in 2013, China sent the then vice governor of the Heilongjiang Province to attend the senior Arctic officials meeting held in Washington, D. C. in 1999 [147]. This reflects China's early interest in the AC. To apply to become an ad hoc observer, China actively hosted an Arctic Science summit week under the International Arctic Science Committee framework in Kunming in 2005 [148] and joined the Ny-Ålesund Science Managers Committee the same year [149] to express its support for the Arctic and to look for and work on common goals and interests [150]. Since being admitted as an ad hoc observer in 2006, China began to engage in the AC Meeting of Senior Arctic Officials [151] and became a member of the planning group members for the International Polar Year in 2007 [152]. This engagement has accumulated experience for China to become an official observer.

After China officially became an AC observer in 2013, it submitted three reports in 2016, 2018, and 2020. On these official recordings, China has put forward proposals in relevant meetings of the working groups, some of which have been adopted by the working groups of the AC. These proposals are either reflected in the plan of the working groups or shown in their reports. In recent years, especially, the number of pertinent proposals adopted by China has increased, which shows the maturity of China's relevant engagement. These proposals involved the formulation of action plans under the Conservation of Arctic Flora and Fauna (CAFF) working group, as well as some evaluation work under the Arctic Monitoring and Assessment Programme (AMAP) working group. These proposals are linked to international law for Arctic governance.

Firstly, climate change can have an effect on the movement of contaminants to the Arctic and their accumulation in the Arctic [153] (p. 1). In some parts of the Arctic, the levels of particular persistent organic pollutants (POPs) are no longer declining to the expected extent, and climate change might be part of the reason [154]. The assessment of Arctic POP-climate interactions is necessary for the joint response to climate change. A researcher from the Chinese Academy of Sciences and two experts participated in the Workshop on POPs and Climate Change which is operated by the Arctic Monitoring & Assessment Program (AMAP) Working Group of the AC in 2019. In the discussion of the assessment report on POPs and climate change, Chinese experts made the proposal to include the reference of POP emissions in the Qinghai-Tibet Plateau and Antarctica, which was accepted in the Workshop [155]. It "fills data gaps and strengthens and supports conclusions" [156] for Arctic POP-climate interaction assessments. The AWAP believes its assessments participated by Chinese experts of POPs in the Arctic "contribute to the arrangements for adding new substances to *United Nations Environment Programme Stockholm Convention on Persistent Organic Pollutants* (the Stockholm Convention) and the *POPs Protocol to the Convention on Long-range Transboundary Pollution* (LRTAP)" and help "evaluate the effectiveness and sufficiency of the Stockholm Convention and LRTAP agreements" [157].

Secondly, although economic activity represents important opportunities for Arctic communities, it also entails environmental challenges that must be handled in the most effective ways possible [158], such as through control of black carbon and methane emissions. A Chinese researcher of the Chinese Research Academy of Environmental Sciences engaged

in the work of ACAP, mainly on the work of black carbon- and methane-related projects and research reports [159]. The report, The *Expert Group on Black Carbon and Methane— third Summary of Progress and Recommendations,* was written as a result of China's active participation. The report calls on all Arctic states to carry out international cooperation in the IMO for a global regulatory framework to reduce black carbon emissions [159]. In response to this report, many sponsors (Canada, Finland, France, Germany, Iceland, Netherlands, Norway, Solomon Islands, Sweden, United Kingdom, and the United States), including Arctic states, submitted a response proposal via the IMO [160]. The proposal finally received the attention of the Committee, and it agreed to further develop the draft MEPC resolution [161].

Finally, because climate change affects the Arctic marine environment, from ecosystem and habitat impacts to driving changes in human activities, marine protected areas are crucial to the resilience of the Arctic as a tool [162]. CAFF's Migratory Bird Work Plan, participated in by Chinese representatives, calls to establish marine protected areas under the Protection of the Arctic Marine Environment (PAME) [163]. Furthermore, the proposal made by China in CAFF's Migratory Bird Work Plan helped establish PAME marine protected areas at the AC level and then led to the output of a new work plan.

The above engagement reveals China's efforts to promote the development of international law involving the paper's theme at the regional level. Before China became an official observer of the AC, China mainly focused on the recognition of the identity of Arctic stakeholders with their own interests and the core position of the AC at the regional level as well as support for the concept and purpose of the AC. Since China did not become an official observer of the AC at that time, the focus of such a national role was mainly to seek the recognition of the AC. Coupled with limited information, it seems difficult to determine that China had a significant implication on the development of international law at the regional level of Arctic governance before China became an official observer of the AC. After China became an official observer of the AC, the main method of China's engagement in the AC was to send experts to the meetings, respect the three agreements adopted by the AC, and support international cooperation through platforms such as the Arctic science and technology ministers' meetings. It is conducive to achieve the AC's sustainable development goals and the SDGs 13–15 related to the Arctic region, because the sustainable development goals at all levels depend on the best scientific evidence, and the relevant scientific data provided by China is advantageous to provide intellectual support for the sustainable development of the Arctic.

As mentioned in Section 3.1 above, the full operation of the core institutions of Arctic governance at the regional level is currently paused. This has delayed the formation of relevant scientific reports, which are the basis for the formulation of international laws and regulations at the Arctic regional level. Accordingly, the temporary pause of the AC prevents China from engaging in the formulation of relevant international laws at the regional level from the perspective of the Arctic governance for its sustainable development.

### 3.3. Implications of China's Arctic Engagement in the International Law Development of Arctic Governance for Its Sustainable Development at the Regional Level

3.3.1. Discussion on China's Role Assessed by Its Official Documents at the Regional Level

At the regional level, China's national role has remained stable. By summarizing the speeches of Chinese leaders at international occasions, the authors find that China maintains the role of supporting and engaging in the AC. Since China first attended the AC Meeting of Senior Arctic Officials in 2007, Chinese representatives have expressed their support of the purposes and objectives of the AC and conveyed their willingness to actively engage in the work of the Council [164]. In 2009, China's then Assistant Foreign Minister expressed his recognition that the AC is the most influential regional intergovernmental organization on Arctic issues and stressed China's desire to engage in the AC work in view of China's own identity and ability [165] (p. 55). In 2010, China issued that "the issue for the AC members now is how to involve non-Arctic states in relevant research endeavours

and discussions at an early stage and in depth" [166]. It represents that China expressed its intention to participate in relevant activities related to the formation of international law at the regional level. Since China became an official observer of the AC in 2015, the then Vice Foreign Minister recognized that Chinese experts actively engaged in scientific research projects of several working groups of the AC [167]. China's 2018 *Arctic policy* reiterates that China fully supports the work of the Council, and dispatches experts to participate in the work of the Council including its Working Groups and Task Forces [27]. In these two official reports, the authors find that the wording adopted by China no longer expresses the desire for active participation but emphasizes that active engagement is a genuine existing circumstances and possible future endeavour.

It is noteworthy that China's diplomatic documents during the pausing of the AC stressed the significance of peace and security to the international community [168] and called for upholding the open regionalism of unity and cooperation [169]. Accordingly, China's role in supporting and engaging in the AC has not changed, regardless of the current operational dilemma of the AC. Although the crisis has objectively hindered China's engagement, it still seeks to support and engage in the AC activities (See Table 3).

**Table 3.** China's Arctic dynamic engagement in the international law development of Arctic governance for its sustainable development at the regional level under the national role.

| Period | International Law Context | China's National Role | Proposals Key Words | Adoption Outcome |
|---|---|---|---|---|
| before the formal establishment of the Arctic Council in 1996 | the formation of regional rules mainly driven by Arctic states and other stakeholders' engagement in the International Arctic Science Committee and the Northern Forum, etc. | an Arctic stakeholder supporting and engaging in the Arctic Council | limited archive as the document acquisition | - |
| after the establishment of the Arctic Council and before the acceptance of China's application for official observer seat in 2013 | the formation of regional rules mainly driven by Arctic states and other stakeholders' engagement in the Arctic Council mechanisms of the senior Arctic officials meeting and the working groups, etc. | | limited archive as the limited authorization by the Arctic Council rules of procedure | - |
| after China officially became the Arctic Council observer in 2013 | | | Arctic migratory bird protection; persistent organic pollutants and climate change | a few adopted proposals assessed under the condition of relatively complete archives |

### 3.3.2. Discussion on China's Arctic Engagement under Its Role at the Regional Level

Before China became an official observer of the AC, China has been given less opportunities to support and engage in the AC. Under limited conditions, China was still willing to engage in the AC by any available opportunity. This phenomenon is illustrated in the consistency of state practice and national role. After China became an official observer of the AC, China has been given more opportunities to support and participate in the AC. Such engagement has mainly given expression to scientists' suggestion to the AC working group's scientific projects.

There is limited room for China to play in the AC compared with the Arctic states at the regional level, although China has made relevant efforts. China cannot carry out more activities in the AC due to its observer status and the AC rules. On account of the rules adopted in 2013, observers can attend meetings of the AC, consult Council documents, and speak at the meeting after statements by states and permanent participants [170], but they

do not have the right to vote. Observers may contribute to the Council's projects and pay for part of the work of the Council, but their financial contributions shall not exceed the funds provided by Arctic states. This means that observers cannot have any substantive impact on the AC system itself. Therefore, within the existing legal framework, China can only act by appointing experts to engage in scientific activities and publishing some literal "observations" [171]. These observer rules objectively limit China's engagement at the regional level. It highlights the necessity for the complementary role of other Arctic bodies outside the AC core of the Arctic governance system at the regional level. It does not contradict the characteristics of China's role at this level, because other Arctic bodies just serve as supplementary fields for China's engagement. Such supplementary fields are in need under the current situation of the AC's comprehensive cooperation pause.

## 4. The Development of International Law and China's Engagement at the Multilateral and Bilateral Levels

### 4.1. The International Law Development of Arctic Governance for Its Sustainable Development at the Multilateral and Bilateral Levels

Activities in bilateral Arctic governance mostly occur within the jurisdiction of Arctic states. Considering that Arctic states and other stakeholders cannot overcome common challenges and threats, respectively, such interdependent relations among Arctic states and other stakeholders require cooperation at comprehensive levels [172,173]. Respect for sovereignty, sovereign rights, and the jurisdiction of Arctic states are prerequisites to launching activities. It is the rational meaning of providing a national framework for integrating development and conservation in the theory of sustainable development that bring together representatives of government, environmental groups, business and industry, indigenous people, and other interests on an equal footing for the establishment of a bilateral and multilateral comprehensive cooperation network. Since cooperation depends on state relations, which are usually more complex, the type of international law on Arctic governance at this level is flexible. It could take the form of hard laws, soft laws, and implied consensus, even with no document, which are the common points of the domestic policies of both sides. This section will examine China's multilateral and bilateral Arctic cooperation based on hard laws, soft laws, and implied consensus.

### 4.2. China's Arctic Multilateral and Bilateral Engagement in Arctic Governance for Its Sustainable Development Based on Hard Law

4.2.1. China's Multilateral Arctic Engagement Based on Hard Law

China's multilateral cooperation based on hard law involves a wide range of examples, and the *Agreement to Prevent Unregulated High Seas Fisheries in the Central Arctic Ocean* (the CAO fisheries agreement) is one of the updated instances of Arctic multilateral cooperation based on hard law. In 2015, the Arctic Five (Russia, United States, Canada, Denmark, and Norway) reached and issued the *Declaration Concerning the Prevention of Unregulated High Seas Fishing in the Central Arctic Ocean*, the predecessor of the CAO fisheries agreement. The declaration states that until there is sufficient scientific evidence to prove the sustainable development of the fisheries in the related area, the Arctic Five will not authorize their vessels to conduct commercial fishing in the stated area. In December of that year, Iceland, China, Japan, the Republic of Korea, and the European Union joined the negotiations on an agreement to restrict fisheries in the abovementioned areas based on the declaration. By the international law-making theory, the authors try to find China's substantive engagement, that is, adopted proposals in negotiation documents related to this agreement, so as to confirm if there is any evidence of China's contribution to the hard law's development for Arctic cooperation at the multilateral level.

At the third meeting of scientific experts on fish stocks in the central Arctic Ocean in May 2015, Chinese representatives delivered a report named *Perspective China Arctic Research Activities—with focusing on the biological information* [174]. Based on the consensus reached by the conference representatives, including China, agreed upon the need to develop a joint programme of research and monitoring, the fourth meeting of scientific

experts on fish stocks in the central Arctic Ocean was successfully held in September 2016 [174]. At the fifth meeting on the same subject in October 2017, Chinese representatives engaged in a monitoring task [175]. In May 2019, the signatory states of the CAO fisheries Agreement established the Provisional Scientific Coordinating Group (PSCG) to further prepare for the implementation of the agreement [176]. China missed the first PSCG meeting hosted in 2020 due to COVID-19-related flight restrictions. However, the meeting document of PSCG pointed out that "a note from the Chinese delegation concerning certain provisions of the proposed Terms of Reference was taken into account during the discussions" [177]. At the first Conference of the Parties to the CAO fisheries agreement held in the Republic of Korea from November 23 to 25, 2022, China, together with other parties, decided to establish a formal scientific coordinating group to carry out research on marine living resources and ecosystems and to strive for the establishment of a joint program of scientific research and monitoring for the central Arctic high seas marine living resources and ecosystems. On account of the international law-making theory, these adopted proposals consolidate the practice of international law based on the best scientific evidence and the precautionary approach and provide a basic legal framework for the sustainable development of fishery resources in the Arctic region. This legal framework is a good beginning for the implementation of SDGs 14 and the AC's sustainable development goal of healthy and resilient Arctic ecosystems.

With China's approval in May 2021 [178], the CAO fisheries agreement has come into effect in June 2021 [179]. In fact, China is the signatory that finally completed the domestic approval. It is necessary to analyse why there was a delay in China's completion of the domestic approval process as well as the reasons for the final completion of the domestic approval. There are several potential reasons. First, stemming from *Law of the People's Republic of China on the Procedure of the Conclusion of Treaties*, the CAO fisheries agreement does not need to be ratified by the Standing Committee of the National People's Congress but only by the State Council [180]. In other words, this is not an agreement in urgent need of approval. Second, in terms of diplomacy, China needs time to fully understand the relevant scientific expertise and develop the necessary diplomacy [181]. Third, there is a great demand for China's pelagic fisheries [182] (p. 99). A ban on fishing activities in pertinent waters of the Arctic Ocean for a vague period of time in the future will increase the pressure on China's domestic fisheries. Despite the above-noted challenges, China has completed the domestic approval procedure, which shows China's determination to promote the achievement of the negotiations.

### 4.2.2. China's Bilateral Arctic Engagement Based on Hard Law

From the perspective of bilateral Arctic cooperation, at present, only Iceland has signed a bilateral intergovernmental agreement on Arctic issues with China, called the *Framework agreement between the government of the People's Republic of China and the government of Iceland on Arctic cooperation* [183]. By the theory of international law-making, the signing result of the agreement denotes that China and Iceland jointly promote the making of bilateral international law. Stemming from this agreement, the two sides are willing to promote the exchange of scientific researchers, explore the establishment of the northern light monitoring station in Iceland, and establish the joint research centre for oceans and polar regions [184]. Policies and programmes for sustainability must be based on scientific knowledge of the factors that they will affect and be affected by. Without a sound basis of scientific knowledge and public understanding of its implications, policies for sustainability are unlikely to be as well formulated or widely supported as they should be. States have to act on the best information they have. The cooperation between China and Iceland in scientific research based on hard law improves the understanding of the Arctic environment and reduces environmental uncertainty that hinders sustainable development. In order to practice the framework agreement, China actively promoted the implementation of it through various cooperation activities, such as the promotion of the exchange of scientific researchers [185] and the construction of the aurora observatory [186]. The abovementioned

activities manifest China's efforts from the perspective of scientific research for the SDGs 14 and the AC's sustainable development goals of healthy and resilient Arctic ecosystems and a healthy Arctic marine environment.

### 4.3. China's Arctic Bilateral Engagement in Arctic Governance for Its Sustainable Development Based on Soft Law

There is a soft law between Finland and China called the *Joint Action Plan 2019–2023* based on a memorandum to carry out joint investigation and research [187]. By the plan document, the two sides agreed to deepen bilateral economic dialogue, promote practical cooperation among enterprises, conduct studies on Arctic-related laws and social science studies and increase knowledge of the Arctic region through science [188]. Based on the soft law, the representatives of China have made many efforts to promote the cooperation plan, such as the joint construction of an icebreaker [189], railway [190], and the research centre [191]. In addition, Russia is another state that reached a soft law on economic cooperation with China on Arctic cooperation based on the "Ice Silk Road" since 2016 [192].

It is worth mentioning that not only China and the Arctic states but also China and other Arctic stakeholders reach soft laws. The Arctic policy released by Japan in 2015, the Arctic policy issued by China in 2018, and the Republic of Korea's *Polar Activities Promotion Act* passed in 2021 all focus on Arctic economic and scientific research activities in the future. Based on this tacit understanding, before the COVID-19 pandemic, China, Japan, and the Republic of Korea held four trilateral high-level dialogues on the Arctic based on the *Joint Declaration for Peace and Cooperation in Northeast Asia* issued in November 2015 and adopted two joint statements. The three Heads of Delegations promoted scientific research as the priority for cooperation among the three countries. They supported the enhancement of the exchange of information on Arctic expeditions and encouraged the sharing of scientific data and further development of collaborative surveys.

By the theory of international law-making, the signing result of the bilateral agreement marks the joint contribution in the making of bilateral international law by Arctic states and other stakeholders. The fields involved in the abovementioned soft law reached between China and Arctic states and other stakeholders is divided into two areas: economic cooperation and scientific research cooperation. The relationship between Arctic scientific research cooperation and Arctic sustainable development has already been analysed in Section 4.2.1. For the Arctic economic cooperation, as mentioned above, every state has a universal goal of economic and social development. The agreements mentioned above on economic cooperation polymerize the universal goals of Arctic states and other stakeholders on sustainable economic development, which is instrumental in meeting the standards for improving the quality of human life in the theory of sustainable development.

There is also another type of cooperation, which is based on implicit consensus reflected by the forms including but not limited to domestic laws and policies. For example, China and Norway decided to increase cooperation on Arctic climate monitoring and prediction in April 2018 [193]. The cooperation has been completed in 2021 based on the platform of the Nansen-Zhu International Research Center [194]. Before the two sides promoted cooperation in 2018, Norway released its Arctic policy in 2006, 2009, 2012, and 2017. These policy contents convey Norway's willingness to launch dialogue with other Arctic stakeholders on some Arctic issues, such as Arctic scientific research and climate change response. China also conveys its aspiration to cooperate with Arctic states on the above Arctic issues in its Arctic policy, which was published in January 2018. The cooperation improves exploration and understanding of the Arctic climate and environmental change and provide scientific support for the realization of the *Paris Agreement* central aims. The realization of the *Paris Agreement* central aims is a manifestation of humanity's pursuit of sustainability. The Arctic region, as an indispensable region in the world, also benefits from the accelerated realization of the *Paris Agreement* central aims resulting from the achievement of the climate change joint response cooperation promoted by China and Norway.

*4.4. Implications of in the International Law Development of Arctic Governance for Its Sustainable Development at the Multilateral and Bilateral Levels*

4.4.1. Discussion on China's Role Assessed by Its Official Documents at the Multilateral and Bilateral Levels

Based on the interdependent relations among Arctic states and other stakeholders mentioned above, China's national role at multilateral and bilateral levels has basically remained stable. It emphasizes cooperation as a keyword of its national role at this level for engaging in the development of international law for Arctic governance (See Table 4). Before the publication of *China's Arctic policy*, China repeatedly stressed respect for the inherent rights of Arctic states and indigenous peoples [195], developing multilevel Arctic cooperation [167,196] and paying attention to the win–win results of Arctic bilateral and multilateral cooperation, especially the commercial use of sea routes [197]. After the publication of *China's Arctic policy*, China continued to adhere to Arctic cooperation. In detail, China states that:

"Cooperation" is an effective means for China's participation in Arctic affairs. It means establishing a relationship of multi-level, omni-dimensional, and wide-ranging cooperation in this area. Through global, regional, multilateral, and bilateral channels, all stakeholders—including States from both inside and outside the Arctic, intergovernmental organizations, and nonstate entities—are encouraged to take part in cooperation on climate change, scientific research, environmental protection, shipping route development, resource utilization, and cultural activities [27].

**Table 4.** China's Arctic dynamic cooperation of Arctic governance for its sustainable development at the multilateral and bilateral levels under the national role.

| Period | International Law Context | China's National Role | The Focus of Cooperation | Characteristics of Cooperation |
|---|---|---|---|---|
| before the publication of China's Arctic policy in 2018 | the enhancement of national win-win results together beneficial to people's welfare of their countries in bilateral and multilateral agreement negotiations between Arctic states and other stakeholders | a partner seeking for Arctic cooperation with Arctic states and other stakeholders | high seas fisheries in the central Arctic Ocean; the promotion of the exchange of scientific researchers; joint marine and polar research; joint construction of icebreaker, railway, and the research centre | mainstream of cooperation based on soft laws, supplemented by cooperation based on hard law; more achievements gained under soft laws compared with hard laws concluded between Arctic states and other stakeholders |
| after the publication of China's Arctic policy in 2018 | | | | |

4.4.2. Discussion on China's Arctic Engagement under Its Role at the Multilateral and Bilateral Levels

In consideration of the requirement of providing a national framework for integrating development and conservation in sustainable development theory, governments should set the creation of a sustainable society as an overall policy goal. To achieve it, they need cooperation in comprehensive fields. China's cooperation on Arctic scientific research, environmental protection, the climate change joint response, and resource development, based on China's role at the multilateral and bilateral levels, is instrumental in the achievement of all sustainable development goals related to the Arctic region.

Among China's cooperation types under the guidance of such a national role, cooperation based on implied consensus is the loosest. It has no independent cooperation document with the set form and frequency. Cooperation based on hard law has the strongest legal effect. Their conclusion is often based on detailed considerations. A strong legal framework is conducive to solving complex issues, including the regional management of fisheries, which calls for solid collective cooperation. It is also instrumental in ensuring the maxi-

mum implementation of cooperation, which is in favour of covering the urgent demand of states. However, the procedures for the formulation of a hard law are cumbersome. The soft laws facilitate this issue. The soft laws are flexible compared with the hard laws and enable participants to carry out activities under recorded conditions selected by both parties compared with the implied consensus.

For the future, China and Arctic states have broader prospects for cooperation. As most Arctic states and other stakeholders have made carbon reduction commitments in response to climate change at the United Nations Climate Change Conference [198], it is valuable to expect bilateral and multilateral emission reduction cooperation for the Arctic. After all, climate change will not be paused due to changes in the regional situation, which is a long-term and consensus task for all states.

### 5. Conclusions

The paper summarizes China's national role and engagement in the development of international law related to the Arctic governance for its sustainable development and its implications. The authors find that under China's national roles mentioned above, China's engagement has affected the development of international law around the mentioned theme to some extent. The summary and conclusions are as follows.

As far as the development of international law related to this theme is concerned, the international law at the global level involves marine studies, Arctic navigation, joint response to climate change, and biodiversity conservation. The international law at the regional level refers to a series of international agreements and public goods adopted under the auspices of the AC. The international legal norms at the multilateral and bilateral levels are the hard laws, soft laws and implied consensus for cooperation. They are closely related to the theory of sustainable development.

In terms of China's national role and engagement, at the global level, China's national role has experienced development from a developing state that needs the mutual assistance of the third world and the care of developed states to then play a role in diplomacy in view of its ability and finally to actively participate in the building of the global governance system. At the regional level, China maintains the role of supporting and engaging in the AC. At the bilateral and multilateral levels, China emphasizes cooperation as a keyword of national role. Based on the abovementioned national roles, China has continuously engaged in the development of international law at all levels and made substantial contributions. These contributions promoted the realization of sustainable development goals at the United Nations and the AC levels.

In order to comprehensively achieve the commitments of *China's Arctic policy* in the future, at the global level, there is a need for China to further optimize the agenda setting to reach more consensus and to "fulfil all its international obligations in accordance with the law" [27]. At the regional level, it is necessary for China to continuously expand the size of the expert group, make good use of limited conditions, and increase the quantity of effective proposals in each working group to achieve "full support" and "high value" [27]. At the bilateral and multilateral levels, the vitality of bilateral and multilateral cooperation based on hard law and soft laws needs to be stimulated so as to achieve the goal of "promoting practical cooperation in all fields" [27]. Inasmuch as the above prediction of China's practice based on *China's Arctic policy*, the authors believe that China does not advocate for breaking the existing international legal order of Arctic governance for its sustainable development, especially in the context of the situation in Ukraine. The conclusion could, to some extent, alleviate doubts of some Arctic states about China's engagement in Arctic affairs related to the theme.

Currently, the international law development of Arctic governance for its sustainable development is facing some challenges, especially the hindrance of the situation in Ukraine for the operation of the AC. Considering the rapidly changing situation in Ukraine, the cooperation based on soft laws and implied consensus reflected by the forms including but not limited to hard laws between Arctic states and non-Arctic states seems to be more

efficient and effective. Such cooperation could be based on the Arctic forums, including but not limited to the AC, or could be organized by the cooperating states themselves. Given that most Arctic states take the issue of climate change response into consideration, the cooperation between Arctic states and non-Arctic states based on clean energy is expected.

**Author Contributions:** Conceptualization, J.B.; methodology, J.B. and K.Z.; writing—original draft preparation, J.B.; writing—review and editing, J.B. and K.Z.; supervision, J.B.; funding acquisition, J.B. All authors have read and agreed to the published version of the manuscript.

**Funding:** This paper is part of a research project funded by China's National Social Sciences Foundation (18VHQ001).

**Institutional Review Board Statement:** Not applicable.

**Informed Consent Statement:** Not applicable.

**Data Availability Statement:** Not applicable.

**Conflicts of Interest:** The authors declare no conflict of interest.

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
