# Peer review of "China’s Engagement in Arctic Governance for Its Sustainable Development Based on International Law Perspective"

_sustainability, doi:10.3390/su15065429_

Round 1

Reviewer 1 Report

This research article was well edited based on authors original research results. However there were so many critical errors. Thus this research article should have to revise before publication as one oe research articles in international journal(s).

Please refer followings

[Major reversion]

1) Title: Please use more simple and easy title.

Ex.: Case study: China's engagement in artic governance for its suitable development based on international law perspective

2] Author's affiliation

Kailei Zhun b

What is this, b? Is it a research article for review process?

Author's affiliation: 1 and 2

3] Objective(s)

Why is this research article important on the view point of globalization not only a specific area or a nation.

Authors need to revise some contents in this research article.

4] Figure(s) and/or Table(s)

If possible, please use Figure(s) and/or Table(s) within 5~7.

Ex.: Map, research flow, etc.

5] Discussion

If possible, please separate some contents from "Results" and then edit as "Discussion".

6] Conclusions

So what? Where and how can we apply based on the results of this research article?

Authors need to consider the applicability of this research article and then revise some contents in "Conclusions".

[Minir reversion]

8] Keywors

Please use only important terms at least 5 terms for "Keywords".

9] References

- Please use reasonable numbers of references

- Please use recent references wihin 5 years.

- Please add all references' publication year.

(1, 14, 15, 16, 17, 18, 24, 27, 43, 46, 56, 60, 73, 76, 78, 79, 80, 81, 85, 86, 88, 97, 07, 108, 115, 118, 119 etc.)

Reviewer 2 Report

Thank you for the opportunity to read the article.

The international presence in the Arctic is a frequently discussed topic, but the article contains many shortcomings, which makes it impossible to evaluate it as a scientific article, and not as a journalistic work.

In line 389, the authors point out that China is constantly putting forward new emission reduction targets. This statement should be detailed and confirmed by sources, and it is also necessary to use arguments to create confidence that these ambitious plans will be fulfilled.

In line 472, the authors suggest that cooperation in the Arctic could take place through the exchange of technologies, which contradicts the established economic relations between business entities of different countries, because they are competitors and it is impossible to ignore the laws of economics in the Arctic or in any other region.

The mechanism of China's influence in matters of biodiversity conservation in the Arctic has not been disclosed at all

In line 737, the author is told that the proposals of Chinese specialists were very effective. This judgment should be supported by facts or references to studies that have evaluated effectiveness.

The article is mainly devoted to highlighting the issues of China's participation in the preparation of various international environmental regulations. Perhaps the authors look at this problem exclusively as specialists in the field of law, but one cannot ignore the many facts about the enormous damage to the environment that take place in China. In particular, there is such a large-scale desertification in Northern China that it also damages neighboring countries. Air pollution, acid rain, soil erosion, pollution of surface and even groundwater, and many other environmental problems remain unresolved in China (https://cyberleninka.ru/article/n/ekologicheskie-problemy-kitaya-2 ). In addition, an accident at a chemical plant in 2005 led to severe pollution of the Songua River, which led not only to the fact that many Chinese were deprived of access to drinking water, but also to the fact that residents of Russia also felt the consequences of this incident, as the Songua flows into the Amur (White Paper "Environmental Protection environments in China. 1996-2005». Press Office of the State Council of the People's Republic of China. 05.06.2006.).

The authors' arguments that China's participation in the development of the Arctic will serve the goals of sustainable development have no basis. In order to assert this, it is necessary to prove that China successfully solves environmental problems on its territory, achieving sustainable development. Without such practice, it is impossible to predict the positive impact of China on the development of the Arctic.

The article has no structure and logic, currently it is difficult to call it a scientific article, methods are not specified, results are not described. The article can be accepted for publication only after serious revision.

Reviewer 3 Report

This paper provides a complete summary of China’s involvement regarding sustainability in the Arctic. Although this paper has a quite significant contribution, the content should be improved before it can be published. Major revisions are suggested for this paper as pointed out below.

Comments

·       Abstract

o   The end of the abstract (starting from line 16) is too general and does not cover the main points explained in the text. The authors should explain some highlights about China’s important role in the Arctic to convince the reader of its significant position.

·       Introduction

o   This paper focused on China’s international policy. But it is also interesting to discuss other countries involved in the Arctic as a constellation and collaboration among different countries.

o   Also explain the role of the arctic region in the global climate system. What is the most possible result of unsustainable development in the Arctic to the global climate?

·       Contents

o   L 203: what does IMO stand for?

o   The discussion is too long, create several figures to illustrate and summarize the discussion at some points, for example, China’s dynamic engagement by period and a table that summarizes China’s regulations.

·       Conclusion

o   The conclusion is just a summary that is contained in previous sections. Give some insight into the future challenges and how China can involve. Also, suggest China’s strategic plan to overcome those challenges.

Round 2

Reviewer 1 Report

I think that this revised research article may be suitable for the publication as one of research articles for international journal(s).

Reviewer 2 Report

Good afternoon. Thank you for the opportunity to read the new version of the article. Thank you also for the answers to my questions and comments. Obviously, the authors have made major changes to the content of the article. Nevertheless, there remains a problem related to the general style of the article. The article does not specify the research methods and methodology followed by the authors. If the methodology had been developed, there would have been a clear line of justification for China's role in sustainable development. For example, in comparison with other countries. Criteria for the effectiveness of this participation would be highlighted. In addition, the list of references contains many references to international agreements, communiques, resolutions. The scientific article should contain links to scientific articles. The use of source No. 5 does not seem to me to be fully justified. Undoubtedly, Sigmund Freud made a great contribution to science, but it is difficult to call him the founder of the doctrine of sustainable development. In this place it is better to refer to the works of V.I. Vernadsky The article needs to be finalized

Reviewer 3 Report

The paper has significantly improved and can be accepted at the present form.

Round 3

Reviewer 2 Report

Thank you for the opportunity to read the new version of the article. Indeed, quoting V.I. Vernadsky is more relevant in this article.